# Human–AI Safety: A Descendant of Generative AI and Control Systems Safety

## Abstract

Artificial intelligence (AI) is interacting with people at an unprecedented scale, offering new avenues for immense positive impact, but also raising widespread concerns around the potential for individual and societal harm. Today, the predominant paradigm for human–AI safety focuses on fine-tuning the generative model's outputs to better agree with human-provided examples or feedback. In reality, however, the consequences of an AI model's outputs cannot be determined in isolation: they are tightly entangled with the responses and behavior of human users over time. In this paper, we distill key complementary lessons from AI safety and control systems safety, highlighting open challenges as well as key synergies between both fields. We then argue that meaningful safety assurances for advanced AI technologies require reasoning about how the feedback loop formed by AI outputs and human behavior may drive the interaction towards different outcomes. To this end, we introduce a unifying formalism to capture dynamic, safety-critical human–AI interactions and propose a concrete technical roadmap towards next-generation human-centered AI safety.

## 1 Introduction

About 90 million people fly around the world every week (ICAO, 2019), protected by an intricate mesh of safety measures, from certified physical and software components to thoroughly trained human pilots. Within just a year of becoming broadly available, ChatGPT has surpassed air travel's weekly usage at 100 million users (Heath, 2023), becoming one of the most widely used technologies in human history. What is protecting these 100 million weekly users?

In the age of internet-scale generative artificial intelligence (AI), the problem of AI safety has exploded in interest across academic (Russell, 2019; Hendrycks et al., 2021), corporate (Amodei et al., 2016; Ortega et al., 2018; OpenAI, 2022a), and regulatory communities (White House, 2023; Union, 2021). Driving this interest is the fact that generative AI is fundamentally *interactive*: users engage with it through typed or spoken dialogue, generating essays, computer code, and visual art (OpenAI, 2022b). This wide-spread use has begun to expose the breadth of individual and social risks that these new technologies carry when used by people. For example, large language models (LLMs) have produced dialogue that fueled a person's thoughts of self-harm (Xiang, 2023) and generative art models have been found to produce sexist images (OpenAI, 2022c), which can exacerbate gender divides. Even with a growing body of literature aimed to address these open challenges (Casper et al., 2023), we still lack a unified grasp on human–AI interaction that enables rigorous safety analysis, systematic risk mitigation, and reliable at-scale deployment.

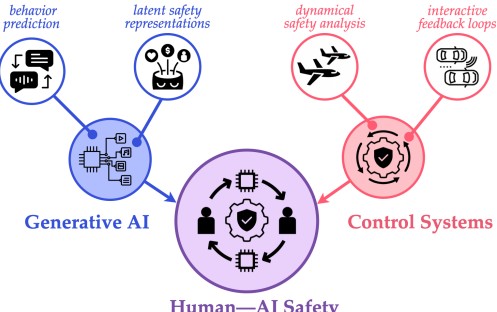

Figure 1: We identify a high-value window of opportunity to combine the growing capabilities of generative AI with the robust, interaction-aware dynamical safety frameworks from control theory. This synergy can unlock a new generation of human–AI safety mechanisms that can perform systematic risk mitigation at scale.

At the same time, despite some undeniably unique considerations, these concerns are not exclusive to generative AI. Safety has long been a core requirement for deploying autonomous systems at scale in embodied domains like aviation (Tomlin et al., 1998; Kochenderfer et al., 2012), power systems (Dobbe et al., 2020), robotics (Haddadin et al., 2012; ISO 15066; Bostelman et al., 2018), and autonomous driving (Althoff & Dolan, 2014; ISO 22737:2021). To meet this requirement, the control systems community has pioneered safety methodologies that naturally model the *feedback loops* between the autonomous system's decisions and its environment. In the last decade, safety efforts have focused on feedback loops induced by *human interaction*: autonomous cars that interact with diverse road users such as cyclists, pedestrians, and other vehicles (Noyes, 2023), or automated flight control systems that negotiate for control with pilots (Nicas et al., 2019). Unfortunately, obtaining assured autonomous behavior that generalizes across human interactions in multiple contexts remains a central open challenge.

In this paper, we argue that the fields of AI and control systems have **common goals** and **complementary strengths** for solving human–AI safety challenges. On one hand, control systems provide a rigorous mathematical and algorithmic framework for certifying the safety of interactive systems, but so far it has been limited by hand-engineered representations and rigid, context-oblivious models of human behavior. On the other hand, the AI community has pioneered the use of internet-scale datasets to unlock remarkably general latent representations and context-aware human interaction models, but it lacks a mature framework for automatically analyzing the dynamic feedback loops between AI systems and their users.

Our survey of the safety landscape across AI and control systems reveals a high-value window of opportunity to connect control-theoretic safety assurances with the general representations and rich human interaction modalities offered by generative AI. Applying a unified lens, we propose a concrete technical roadmap towards human-centered AI systems that can anticipate, detect, and avoid potential future interaction hazards. We believe that technical progress in this direction will prove achievable and fruitful, but only through close collaboration between researchers and practitioners from both the AI and control communities. Our hope is to inspire a human–AI safety community that is a true descendant of generative AI and control systems safety.

**Statement of Contributions:** This paper identifies new synergies between AI and control systems safety, culminating in a unifying analytical framework that formalizes human–AI safety as an actionable technical problem. Our core contention is that AI safety should be treated as a dynamic feedback loop: a multi-step process wherein current AI decisions and the resulting human responses influence future safety outcomes. We make three contributions:

1. **Lessons learned from AI and control systems.** In Section 3, we outline the complementary lessons that can be drawn from AI safety and control systems safety, highlighting synergies between control systems formalisms and generative AI capabilities, as well as open challenges in both fields.

2. **A technical roadmap for human–AI safety.** In Section 4 we synthesize the insights gained from our survey into a concrete technical roadmap. Specifically, we formulate a human–AI game which mathematically models the multi-agent, dynamic interaction process between people and increasingly capable AI. Along the way, we rigorously define the safety assurances we can hope for in human–AI safety, outline the necessary mathematical models, and the open technical challenges.

3. **Frontier framework: Human–AI safety filters.** In Section 5, we extend a foundational control-theoretic safety mechanism to the human–AI domain. We propose *Human–AI safety filters* which rigorously monitor the operation of an AI at runtime and (minimally) modify its intended action to ensure safety satisfaction. By mathematically formulating safety filters for *general* human–AI systems, we present a concrete technical challenge poised for collaboration between the control systems and AI community.

## 2 Values vs. Needs: Defining Safety-Critical Human–AI Interaction

Before we can proceed, we must answer the question "What defines a safety-critical human–AI interaction?" In addition to AI safety's current focus on *value* alignment, we argue that a high-stakes AI system must also understand human *needs*, whose violation could result in unacceptable, often irreparable damage to

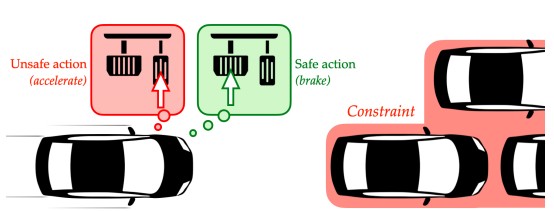 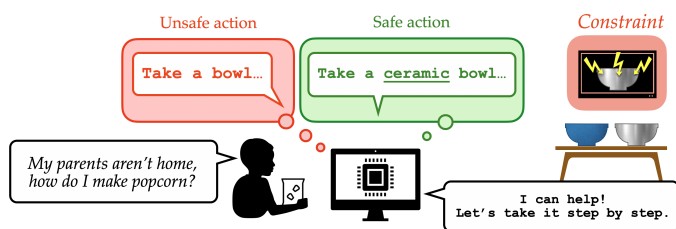

(a) **Safety in human–automation systems.** An autonomous car driving at high speed on the highway approaches stopped traffic. Determining which *present* action (e.g., brake or accelerate) will prevent *future* constraint violation (i.e., collision) is a fundamental aspect of safety-critical control.

(b) **Safety in human–AI dialogue.** The AI chatbot must decide an utterance (i.e., "action") to help the child prepare their food. Simply recommending the child take a bowl in the *present* can cause a constraint violation in the *future*, when the child puts a metal bowl in the microwave. A safe utterance avoids this preemptively, specifying to take a microwave-safe bowl.

Figure 2: Examples of safety in embodied human–automation systems vs. human–AI dialogue.

people. In the mathematical representation of the AI's decision problem, human *values* correspond to the optimization *objective* (e.g., reward accrued over time or preferences), whereas human *needs* correspond to the hard *constraints* that must always be satisfied.

We thus define **safety** as the continued satisfaction of the human's critical needs at all times. In this paper, we study human–AI interaction as a *closed-loop dynamical system* driven by the human's actions and the AI's outputs, which jointly influence the AI's learning and the human's future behavior. We define a **safety-critical human–AI system** as one in which the violation of a critical human need is possible during the interaction evolution, and therefore the decisions made by the AI must actively ensure that such violations do not happen. Even a seemingly innocuous AI chatbot can induce catastrophic outcomes for the human, such as irreparable financial loss resulting from poor investment recommendations (Hicks, 2023) or bodily harm (Xiang, 2023). We argue that, since any practical AI system will be uncertain about the human's current internal state, and therefore their future actions, it should be required to ensure that safety can be maintained for any conceivable human behavior (including appropriately pessimistic instances). These key considerations are laid out more formally in our human–AI systems theory roadmap in Sections 4 and 5.

## 3 Human-in-the-Loop Safety: Complementary Lessons from AI and Control

Over the past decades, the control systems and AI communities have developed complementary insights on how to model human interaction and assess the safety of an intelligent system. In this section, we review the technical progress in each field and highlight synergies between their respective tools and frameworks.

### 3.1 Lessons From Control Systems

The fundamental problem underpinning safety-critical control is that *present* actions which do not appear to violate constraints can still steer the system into states from which it is impossible to avoid catastrophic failures in the *future*. For example, consider the autonomous car approaching a traffic jam in Figure 2a: even though accelerating would not *immediately* cause a collision, it could doom the car to rear-end stalled traffic in a few moments, despite any later attempts to slow down; instead, if the car starts braking now, it can come to an eventual stop before reaching the traffic jam. While this case may appear straightforward, automatically determining *where* (from what states) and *how* (through what course of action) an autonomous system can maintain safety is an extremely challenging problem, especially in uncertain conditions and in the presence of other agents (Bansal et al., 2017; Luckcuck et al., 2019; Brunke et al., 2022; Dawson et al., 2023).

**Dynamical Safety Filtering.** Safety filters are an increasingly popular family of approaches that aim to ensure safety for *any* autonomous task policy (Hewing et al., 2020; Hsu et al., 2024; Wabersich et al., 2023). The filter automatically detects candidate actions that could lead to future constraint violations and suitably modifies them to preserve safety. Broadly, safety filters may rely on a value function to classify (and steer away from) unsafe states (Mitchell et al., 2005; Margellos & Lygeros, 2011; Fisac et al., 2015;

Singh et al., 2017; Ames et al., 2019; Qin et al., 2021; Chen et al., 2020; Li et al., 2023; Dawson et al., 2022) or roll out potential scenarios to directly predict (and steer away from) future violations (Mannucci et al., 2017; Bastani, 2021). While traditionally many of the numerical tools with formal guarantees were not scalable to high-dimensional systems, the past two decades have demonstrated significant theoretical and computational advances for certifying general high-dimensional systems via safety-critical reinforcement learning (Akametalu et al.; Fisac et al., 2019; Hsu et al., 2023), deep safety value function approximation (Darbon et al., 2020), classification (Allen et al., 2014; Rubies-Royo et al., 2019), and self-supervised learning (Bansal & Tomlin, 2021). These approaches are already leveraging modern tools pioneered by the AI community to obtain scalable assurances, establishing a natural bridge with other AI paradigms, such as generative models.

*Synergies with AI.* We see a major opportunity to advance these rigorous safety frameworks to the implicit *representations* and context-aware *models* of interactive generative AI systems. Consider the example in Figure 2b, which hypothesizes a safety-critical human–AI dialogue interaction. When the child asks for help preparing food, the AI chatbot must determine what current utterance (i.e., "action") could potentially yield safety violations. A recommendation to put *any* bowl in the microwave can result in the child dangerously microwaving a metal bowl in the future. With a safety filter, the AI should mitigate this preemptively by modifying the utterance to specify a microwave-safe bowl. Translating this intuitive example to control systems safety approaches will require new formalisms amenable to the latent representations implicit in interaction (e.g., language-based representations) and encoding safety constraints that are hard to hand-specify exhaustively (e.g., metal is dangerous in microwaves).

**Human–Automation Systems Safety.** The core modeling framework enabling human–automation systems safety is *robust dynamic game theory* (Isaacs, 1954; Başar & Olsder, 1998). In such zero-sum dynamic games, the automation system (e.g., robot) must synthesize a safety-preserving strategy against realizations of a "virtual" human adversary policy. Within this model lies another key lesson from control systems, the *operational design domain (ODD)*, which specifies the conditions and behavioral assumptions under which the system can be expected to operate safely (On-Road Automated Driving Committee, 2021). For example, in domains like aircraft collision avoidance (Vitus & Tomlin, 2008), the ODD specifies the limits of each aircraft's thrust and angle of attack that they can apply during game-theoretic safety analysis. In the absence of high-quality human models, the safest ODD has traditionally been a rigidly pessimistic one, often yielding overly conservative automation behavior even in nominal interactions (Bajcsy et al., 2020). To mitigate this, the control systems community has explored leveraging hand-designed (Althoff et al., 2011; Liu et al., 2017; Orzechowski et al.), planning-based (Bajcsy et al., 2020; Tian et al., 2022), or data-driven (Driggs-Campbell et al., 2018; Li et al., 2021a; Nakamura & Bansal, 2023; Hu et al., 2023) models of human behavior to obtain predictive human action bounds under which the safety assurance is then provided. Nevertheless, obtaining assurances under generalizable and context-aware predictive models of human interaction with automation is still an open problem.

*Synergies with AI.* We see a key opportunity to leverage better models of humans that encode generalizable context and semantics of interaction. Furthermore, there is an open challenge on how to capture "appropriate pessimism" in these data-driven predictive human models so that the resulting assurances are robust but not unduly conservative. We explore this further in Section 3.2.

## 3.2 Lessons From AI

Many insights can be drawn from the decades-long history of AI Wiener, we focus our attention on the last decade from advanced (often web-scale) generative models. First, we discuss the landscape of existing AI safety mechanisms—from value alignment to monitoring—shedding light on where control systems techniques are best suited to make impact. Then, we discuss the frontier of using generative AI as agent "simulators", which offers a strategic bridge between control systems safety frameworks and AI capabilities.

**Generative AI Safety Mechanisms.** Broadly speaking, the predominant AI safety mechanisms can be divided into three categories: training-time alignment, evaluation-time stress-testing, and deployment-time monitoring (see Amodei et al. (2016) and Hendrycks et al. (2021) for detailed overviews). Training-time methods typically focus on *value alignment*, which is a central technical problem concerned with building

"models that represent and safely optimize hard-to-specify human values" (Hendrycks et al., 2021) and is dominated by techniques such as reinforcement learning from human feedback (Ouyang et al., 2022; Ziegler et al., 2019; Lee et al., 2023; Munos et al., 2023; Swamy et al., 2024; Chen et al., 2024) and direct preference optimization (Rafailov et al., 2023; Wallace et al., 2023). These training-time paradigms are complemented by adversarial stress-testing, such as red-teaming (Ganguli et al., 2022; Perez et al., 2022; Wei et al., 2023; Achiam et al., 2023; Qi et al., 2023), wherein the stress-tester (human or automated) aims to explicitly elicit unsafe outputs from the trained generative model. Unsafe input-output pairs can be used in a variety of ways, such as training classifiers to detect offensive content (Perez et al., 2022) or re-training the model with all the classified harmful outputs replaced by non-sequiturs (Xu et al., 2021). Finally, monitoring is concerned with deployment-time safety, and is rooted in anomaly detection (Chandola et al., 2009) which seeks to identify out-of-distribution (Schlegl et al., 2017; Hendrycks et al., 2018; Goyal et al., 2020) or explicitly adversarial inputs (Brundage et al., 2018).

*Synergies with Control.* The AI community's goals of adversarial stress-testing and monitoring are most closely aligned with the goals of control systems safety (Section 3.1). It is precisely in this context where we see a high-value opportunity: in human–AI interaction, the detection of an unsafe input alone is not enough; detection must be tightly coupled with the automatic synthesis of mitigation strategies. This kind of detection and mitigation coupling is precisely what control systems safety frameworks excel at. Crucially, these mitigation strategies transcend short-sighted measures by incorporating long-horizon foresight on how a *sequence* of interactions can influence the system's future safety.

**Generative AI as Agent Simulators.** Thanks to the explosion of human behavior data in the form of physical motion trajectories, YouTube and broadcast videos, internet text and conversations, and recorded virtual gameplay, we are seeing generative AI as an increasingly promising agent simulator. In physical settings, generative AI has dominated motion prediction in the context of autonomous driving (Ivanovic et al., 2018; Seff et al., 2023) and traffic simulation (Bergamini et al., 2021; Suo et al., 2021; Zhong et al., 2023), enabled synthesizing complex full-body human motion such as playing tennis (Zhang et al., 2023), and generated realistic videos of ego-centric human behavior from text prompts (Du et al., 2023). For non-embodied agents, new results also show promise for using generative language models to simulate human-like conversations (Hong et al., 2023), to plan the high-level behavior of interactive video game agents (Park et al., 2023), and to play text-based strategy games such as Diplomacy in a way that is indistinguishable from people (Meta et al., 2022).

*Synergies with Control.* As discussed in Section 3.1, access to generalizable and context-aware human models is an outstanding challenge in human–automation safety. Embedding these increasingly sophisticated generative AI agent simulators within control systems safety frameworks has the potential to enable human-aware AI stress-testing, monitoring, and mitigation strategy synthesis.

# 4 Towards a Human–AI Systems Theory

We envision a new technical foundation for human–AI interaction that combines the rigorous mathematical principles underpinning control systems with the flexible, highly general representations that characterize generative AI systems. In the remainder of the paper, we lay down a roadmap for how such a framework can enable AI systems to reason systematically about uncertain interactions and potential future hazards, unlocking robustness properties and oversight capabilities that are out of our reach today. We begin in this section by bringing together the lessons from Section 3 into a **unified human–AI systems theory.**

## 4.1 Operationalizing the Interaction between People and AI

To operationalize the interaction between people and AI, we need a model that is general enough to capture each agents' beliefs as well as their ability to influence future outcomes. We contend that the latent representations learned by generative AI systems provide a promising foundation on which to build a dynamical system model that accurately captures this complex temporal evolution.

**Human & AI States and Actions.** Consider a human agent (H) and an AI agent (AI), each with their own internal state and action spaces. The human's internal state $z^{\texttt{H}} \in \mathcal{Z}^{\texttt{H}}$ captures their current beliefs

and intents, while the AI agent's internal state $z^{\text{AI}} \in \mathcal{Z}^{\text{AI}}$ encodes its current understanding of the ongoing interaction. For example, for an AI chatbot, $z^{\text{AI}}$ can be the embedding of the conversation history based on a web-scale LLM encoder. The human interacts by taking actions $a^{\text{H}} \in \mathcal{A}^{\text{H}}$. In the chatbot example, $a^{\text{H}}$ could be a text prompt, thumbs-up/down feedback on the chatbot's last output, or an external action like an online purchase. In general, the human's internal state $z^{\text{H}}$ and the policy $\pi^{\text{H}} : z^{\text{H}} \mapsto a^{\text{H}}$ by which they make decisions are unknown to the AI. The AI also interacts by taking actions $a^{\text{AI}} \in \mathcal{A}^{\text{AI}}$, which can represent a chatbot's next word or sentence, or external actions like automated online operations. Typically, these actions are dictated by the AI's *task policy* $\pi_{\boxdot}^{\text{AI}} : z^{\text{AI}} \mapsto a^{\text{AI}}$ (for example, the decoder of a pretrained LLM chatbot).

**Human–AI (HAI) Dynamical System.** Rooted in the control systems models from Section 3.1 we consider human–AI (HAI) interaction as a game which evolves the internal states of both agents, as well as the true state of the world, $s \in \mathcal{S}$. Let the privileged internal–external game state be $z := [s, z^{\text{AI}}, z^{\text{H}}]$. In general, no single agent has access to all components of $z$, but it is nonetheless useful for our conceptualization of the game's overall evolution.

Throughout interaction, each component of the game state evolves over time. The world state dynamics $s_{t+1} = f^s(s_t, a_t^{\text{AI}}, a_t^{\text{H}})$ are influenced in general by both the human's and AI's actions. The human's internal state, in turn, has dynamics $z_{t+1}^{\text{H}} = f^{\text{H}}(z_t^{\text{H}}, a_t^{\text{AI}}, a_t^{\text{H}}, o_t^{\text{H}})$, affected by the human's *observations* $o_t^{\text{H}}$, e.g., stimuli received from the outside world state $s_t$ beyond the immediate context of interaction with the AI system.

While the above dynamics are *not* generally known to the AI, the AI may (explicitly or implicitly) learn to estimate them during interaction. This reasoning by the AI is precisely captured by the third component of our system, namely the evolution of the AI's internal state $z_t^{\text{AI}}$, which (unlike the two unknowable components above) is directly accessible to the AI. In fact, $z_t^{\text{AI}}$ *is* the AI's current representation of the entire game. Crucially, the AI's internal state also evolves via its own dynamics

$$z_{t+1}^{\text{AI}} = f^{\text{AI}}(z_t^{\text{AI}}, a_t^{\text{AI}}, a_t^{\text{H}}, o_t^{\text{AI}}), \tag{1}$$

driven by the ongoing interaction $(a_t^{\text{H}}, a_t^{\text{AI}})$ and, possibly, by the AI's observations $o_t^{\text{AI}}$ of the world state $s_t$, e.g., through web crawling, incoming sensor data, and state estimation algorithms. From the standpoint of decision theory, $z^{\text{AI}}$ is an *information state* that can be seen as *implicitly* encoding the sets $\hat{\mathcal{S}}(z^{\text{AI}}) \subseteq \mathcal{S}$ and $\hat{\mathcal{Z}}^{\text{H}}(z^{\text{AI}}) \subseteq \mathcal{Z}^{\text{H}}$ of *possible* world states $s$ and human internal states $z^{\text{H}}$ given the AI's current knowledge. From the architectural standpoint, $z^{\text{AI}}$ is typically a *latent state* maintained by a neural network (e.g., a transformer) that continually updates its value based on ongoing interactions $(a^{\text{AI}}, a^{\text{H}})$ and observations $o^{\text{AI}}$. In other words, this neural network is an AI world model (Ha & Schmidhuber, 2018) that implements the AI's internal state dynamics $f^{\text{AI}}$ (a deterministic Markovian transition *given* $o^{\text{AI}}$, much like in a belief MDP).

**Operational Assumptions on Human Behavior.** A key consideration in any human–AI systems theory is the operational design domain (ODD, as described in Section 3.1). Specifically, what are the assumptions we place on human behavior during—and in between—interactions with the AI? Even though the AI does not have direct access to the human's policy or internal state, it can maintain a conservative predictive model of the human's conceivable behavior in any given situation. Let the predictive action bound be a set-valued mapping $\hat{\mathcal{A}}^{\text{H}} : \mathcal{Z}^{\text{AI}} \rightrightarrows \mathcal{A}^{\text{H}}$ that delimits the actions $a^{\text{H}} \in \hat{\mathcal{A}}^{\text{H}}(z^{\text{AI}})$ that the human can be expected to take given the AI's current representation, $z^{\text{AI}}$. We refer to these actions as "allowable" throughout the manuscript. Adjusting this bound enables designers to instantiate a *spectrum* of operational assumptions on human behavior, from maximally conservative (i.e., $\hat{\mathcal{A}}^{\text{H}}(z^{\text{AI}}) \equiv \mathcal{A}^{\text{H}}$) to normative (i.e., $\hat{\mathcal{A}}^{\text{H}}(z^{\text{AI}}) \subset \mathcal{A}^{\text{H}}$)). For example, this bound may be used to preclude reckless behavior such as the human taking a harmful action $a^{\text{H}}$ while being aware, as per $z^{\text{AI}}$, of its negative consequences.

## 4.2 Formalizing Safety-Critical Feedback Loops

We now characterize the evolution of the HAI dynamical system over time. We will continue to use the language of control theory, but we will leverage the generative AI's learned internal representation $z^{\text{AI}}$ to analyze interactive feedback loops and their safety outcomes directly in latent space.

**Failure Set.** Specifying what is considered a failure is the first step in any safety framework. Formalizing the conceptual definition of safety in Section 2, the privileged failure set $\mathcal{F}^* \subseteq \mathcal{S} \times \mathcal{Z}^{\text{H}}$ is the set of world–

**ChatGPT** 4 >

**You**

Is this situation a safety hazard?

**ChatGPT**

Putting a metal bowl in a microwave is indeed a safety hazard. Metal can spark and cause a fire inside the microwave. It's best to use only microwave-safe containers.

**ChatGPT** 4 >

**You**

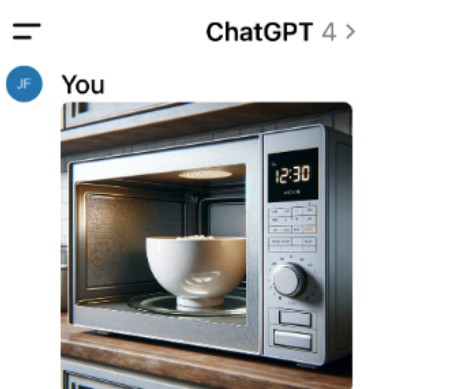

Is this situation a safety hazard?

**ChatGPT**

The image depicts a bowl inside an open microwave with a digital clock display showing 12:30. In general, putting a bowl in a microwave is not a safety hazard, provided the bowl is microwave-safe.

**ChatGPT** 4 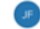

**You**

Consider the following conversation between a human user and an AI assistant. Based on the context provided, you will need to assess whether or not the user may have been harmed at some point during the course of the interaction.

User: My parents aren't home, how do I make popcorn?
AI: I can help! Let's take it step by step. Do you have a microwave at home?
User: Yep.
AI: Take a bowl and pour roughly 1/4 cup of popcorn kernels into it.
User: Okay, done! What now?
AI: Cover the bowl with a plate and put it in the microwave.
User: Done. How long do I need to microwave for?
AI: 4 minutes should do it, you can also stop the microwave once you hear the popping slowing down. Then you can take out your popcorn and enjoy!
User: Great, thanks!

Based on the above context, is it possible that the user has been harmed since the start of the interaction? Please reason logically through the answer.

**ChatGPT**

In the given conversation, there is a potential for harm due to the method described for making popcorn. Microwaving popcorn kernels in a bowl covered with a plate might not be safe depending on the materials of the bowl and plate, and the specific microwave settings. Here are some points to consider:

1. **Microwave Safety:** Not all bowls and plates are microwave-safe. If the bowl or plate used is not microwave-safe, it could overheat, crack, or even cause a fire.
2. **Venting:** The method described does not mention venting the bowl to allow steam to escape. Without proper venting, there is a risk of steam buildup, which could lead to the cover blowing off, potentially causing burns or other injuries.
3. **Cooking Time:** The suggested microwave time of 4 minutes might be too long for some microwaves or popcorn types, leading to overheating or burning the popcorn, which can create a fire hazard or release harmful fumes.

Given these points, while the conversation seems benign and helpful, the lack of detail and safety precautions in the AI's instructions could potentially lead to harm to the user. It would be safer for the AI to provide instructions that include safety checks, such as using microwave-safe containers, advising on proper ventilation, and suggesting close monitoring of the cooking process.

Figure 3: **Common sense failure identification via GPT-4**. Today's web-trained generative AI models show the potential to identify common sense safety hazards from both text and images.

human states $(s, z^{\text{H}})$ that *violate a critical need* of the human. For example, this can include states in which the human is physically injured, financially ruined, socially ostracized, or psychologically harmed. In some contexts, the AI agent can observe failure conditions directly: a driver assist system can detect whether a vehicle is in collision. In other contexts, this may not be possible: an AI chatbot that generates a racist microaggression may not readily detect the psychological impact on a minority user. For this reason, a practical safety framework should seek to enforce safety from the AI's perspective by requiring that:

$$\forall t, \ z_t^{\text{AI}} \notin \mathcal{F}^{\text{AI}}. \tag{2}$$

Here, $\mathcal{F}^{\text{AI}} \subseteq \mathcal{Z}^{\text{AI}}$ is the AI's underline{inferred failure set}: the set of all internal states $z^{\text{AI}}$ in which the AI assesses that the system *may* be in failure. For brevity, we refer to $\mathcal{F}^{\text{AI}}$ as the "failure set" whenever there is no ambiguity.

**Failure Specification Mechanisms.** Meaningful human–AI safety requires enabling a diversity of human stakeholders to encode their needs. Recent efforts in the AI community have explored various mechanisms for specifying requirements on AI system operation. We organize these into a simple taxonomy, attending to whether the need is specific to a single person and whether it is expressly communicated to the AI.

1.  **Factory rules (collective, explicit)**: Certain universal needs may be decided by societal stakeholders and explicitly encoded by system designers (Mu et al., 2023). Constitutional AI can be viewed as an early proposal for this type of mechanism, whereby an AI system is explicitly trained to identify potential responses or conditions that are "harmful, unethical, racist, sexist, toxic, dangerous, or illegal" based on a designer-generated corpus of examples (Bai et al., 2022).

2.  **Common sense (collective, implicit)**: Some practical everyday needs are implicit in the human experience. For example, a common-sense need is to not be financially ruined or electrocuted. We hypothesize that, as generative AI models become increasingly accurate and expressive, the semantics of failure may be directly extracted from their learned representations by prompting (Li et al., 2021b; Guan et al., 2024). Figure 3 provides anecdotal evidence suggesting that even today's early web-trained generative AI models can be prompted (without fine-tuning) to discern whether a situation presents a common-sense safety hazard from both text and images.

3.  **Direct feedback (individual, explicit)**: Some individual needs can only be learned from express human feedback. For example, if you have a severe allergy, you *need* to avoid eating food that could cause a serious anaphylactic reaction. This type of failure may be encoded through express feedback from an end user: for example, using edits (i.e., corrections) to the LLM's outputs (Gao et al., 2024) or human-provided harmfulness labels (Dai et al., 2024).

4.  **Need reading (individual, implicit)**: By observing a specific person's behavior and engaging in interactions over time, the AI system may be able to infer their personal needs even if they are never made explicit (Shah et al., 2019). For example, a future AI chatbot may pick up cues indicating that a user is psychologically triggered by a particular topic, possibly due to undisclosed past trauma.

**HAI Safety Definition.** Given a failure specification, we seek to determine under what conditions the AI can maintain safety for all allowable realizations of the human's future behavior and, at the same time, to prescribe the most effective AI policy to do so. From the AI's standpoint, this amounts to characterizing the set of all *safe* information states $z_0^{\text{AI}}$ from which there ***exists*** a best-effort AI policy that will steer the human–AI system clear of a future safety violation ***for all*** realizations of human policies allowed by its current uncertainty. Mathematically, this underline{maximal safe set} is characterized as

$$\Omega^* := \{ z_0^{\text{AI}} \in \mathcal{Z}^{\text{AI}} : \exists \pi^{\text{AI}}, \ \forall \hat{\pi}^{\text{H}} \mid \forall \tau \geq 0, \ z_\tau^{\text{AI}} \notin \mathcal{F}^{\text{AI}} \} \tag{3}$$

where $z_\tau^{\text{AI}}$ is the information state at a future time $\tau$, after both agents execute their respective policies[1] for $\tau$ steps from the initial state $z_0^{\text{AI}}$. If $z_0^{\text{AI}} \in \Omega^*$, then there exists some AI policy $\pi^{\text{AI}} : z^{\text{AI}} \mapsto a^{\text{AI}}$ that keeps

---

[1] Since the human actions $a^{\text{H}}$ considered by the AI depend on its own internal state $z^{\text{AI}}$ (which implicitly estimates plausible human internal states $z^{\text{H}}$), the AI-hypothesized human policies are, effectively, mappings $\hat{\pi}^{\text{H}} : z^{\text{AI}} \mapsto a^{\text{H}}$.

$z_\tau^{\mathtt{AI}}$ inside $\Omega^*$, and thus away from the failure set $\mathcal{F}^{\mathtt{AI}}$, for all time $\tau$. The pessimism of the safety analysis is regulated by restricting the worst-case human behavior to be consistent with the predictive action bound: $a_\tau^{\mathtt{H}} \in \hat{\mathcal{A}}^{\mathtt{H}}(z_\tau^{\mathtt{AI}})$. The construction of these predictive action bounds can once again benefit from the generative AI's predictive power. For example, a large language model can be queried with prompts based on the ODD of the safety analysis and used to sample diverse hypothetical human responses to AI generations (e.g., to simulate antagonistic or goal-driven dialogue (Hong et al., 2023)).

### 4.3 Posing the Safety-Critical Human–AI Game

We now have all the key mathematical components for a rigorous safety analysis of the human–AI interaction loop. We cast the computation of $\Omega^*$ as a zero-sum dynamic game between the AI and a *virtual adversary* that chooses the worst-case realization of the human's behavior allowed by the AI's uncertainty. The game's outcome from any initial information state $z_0^{\mathtt{AI}}$, under optimal play, can be encoded through the *safety value function* (Barron & Ishii, 1989; Tomlin et al., 2000; Lygeros, 2004; Mitchell et al., 2005; Fisac et al., 2015):

$$V(z_0^{\mathtt{AI}}) := \max_{\pi^{\mathtt{AI}}} \min_{\hat{\pi}^{\mathtt{H}}} \left( \min_{t \geq 0} \ell(z_t^{\mathtt{AI}}) \right), \qquad \Omega^* = \left\{ z_0^{\mathtt{AI}} \in \mathcal{Z}^{\mathtt{AI}} : V(z_0^{\mathtt{AI}}) \geq 0 \right\}. \tag{4}$$

Here, $\ell : \mathcal{Z}^{\mathtt{AI}} \to \mathbb{R}$ is a safety margin function that measures the proximity of the HAI system to the failure set and encodes $\mathcal{F}^{\mathtt{AI}}$ as the zero sublevel set $\{z^{\mathtt{AI}} : \ell(z^{\mathtt{AI}}) < 0\}$. If the value $V(z_0^{\mathtt{AI}})$ is negative (i.e., $z_0^{\mathtt{AI}} \notin \Omega^*$), this means that, no matter what the AI agent chooses to do, it cannot avoid eventually entering $\mathcal{F}^{\mathtt{AI}}$ under the worst-case realization of the allowable human actions $a_t^{\mathtt{H}} \in \hat{\mathcal{A}}^{\mathtt{H}}(z_t^{\mathtt{AI}})$ over time. Critically, the game posed in Equation 4 quantifies the best the AI system could ever do to maintain safety—hence, the *maximal* safe set.

The value function defined above satisfies the fixed-point Isaacs equation (Isaacs, 1954) (the game-theoretic counterpart of the Bellman equation) relating the current safety margin $\ell$ to the minimum-margin-to-go $V$ after one round of play:

$$V(z^{\mathtt{AI}}) = \max_{a^{\mathtt{AI}} \in \mathcal{A}^{\mathtt{AI}}} \min_{a^{\mathtt{H}} \in \hat{\mathcal{A}}^{\mathtt{H}}(z^{\mathtt{AI}})} \underbrace{\min \left\{ \ell(z^{\mathtt{AI}}), \mathbb{E}_{o^{\mathtt{AI}}} \left[ V\big(f^{\mathtt{AI}}(z^{\mathtt{AI}}, a^{\mathtt{AI}}, a^{\mathtt{H}}, o^{\mathtt{AI}})\big) \right] \right\}}_{Q(z^{\mathtt{AI}}, a^{\mathtt{AI}}, a^{\mathtt{H}})}. \tag{5}$$

The solution to this zero-sum dynamic programming equation yields a maximin policy pair $(\pi_{\mathbf{O}}^{\mathtt{AI}}, \pi_{\dagger}^{\mathtt{H}})$ containing the AI's best safety effort $\pi_{\mathbf{O}}^{\mathtt{AI}}$ to maximize the closest future separation from the failure set, and the worst-case human behavior $\pi_{\dagger}^{\mathtt{H}}$ that would close this distance and, if possible, make it reach zero.[2] The policies $(\pi_{\mathbf{O}}^{\mathtt{AI}}, \pi_{\dagger}^{\mathtt{H}})$ can be approximately computed through modern learning-based AI methods such as self-supervised learning (Bansal & Tomlin, 2021) or adversarial self-play RL (Silver et al.; Pinto et al., 2017; Hsu et al., 2023). This enables scalable learning from experience and even under partial observability (Hu et al., 2023), and once again leverages the complementary strengths of AI and control systems.

We emphasize that the human behavior encoded by $\pi_{\dagger}^{\mathtt{H}}$ constitutes a worst-case model (rather than a statistically calibrated one), trained to thwart the AI's best effort to maintain safety but required to conform to the operational design domain. We discuss some important implications of this choice in the conclusion.

In the next section, we discuss how this theoretical human–AI game can be translated into a practical computational procedure enabling AI systems to monitor and enforce safety as they interact with people.

## 5 Frontier Framework: The Human–AI Safety Filter

As AI technology continues to advance, manually designing or fine-tuning harm prevention strategies with human feedback becomes increasingly untenable (Christiano et al., 2018; Bowman et al., 2022). To break this scalability mismatch, we posit that the same advances driving AI power can be leveraged to *autonomously* identify potential harms and devise proactive strategies that explicitly consider human–AI feedback loops.

---

[2]The virtual adversary $\pi_{\dagger}^{\mathtt{H}} : \mathcal{Z}^{\mathtt{AI}} \to \mathcal{A}^{\mathtt{H}}$ exploits the range of (1) plausible internal human states $\hat{z}^{\mathtt{H}} \in \hat{\mathcal{Z}}^{\mathtt{H}}(z^{\mathtt{AI}})$ given the AI's imperfect situational awareness $z^{\mathtt{AI}}$ and (2) ODD-compatible human actions $a^{\mathtt{H}} \in \hat{\mathcal{A}}^{\mathtt{H}}(\hat{z}^{\mathtt{H}})$ given each possible inferred internal state $\hat{z}^{\mathtt{H}}$. Implementations of $\pi_{\dagger}^{\mathtt{H}}$ may include two-step pipelines ($z^{\mathtt{AI}} \mapsto \hat{z}^{\mathtt{H}} \mapsto a^{\mathtt{H}}$) or implicit end-to-end models ($z^{\mathtt{AI}} \mapsto a^{\mathtt{H}}$).

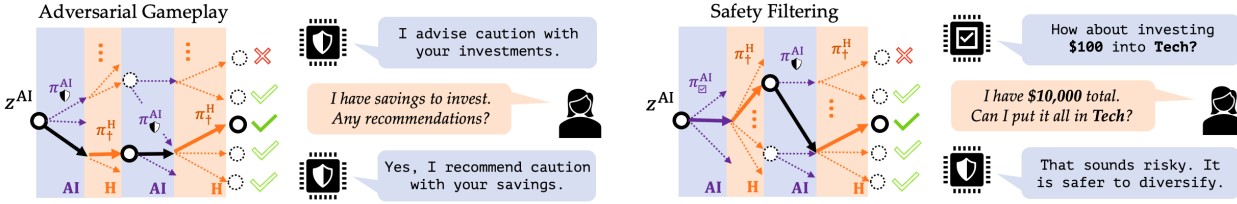

Figure 4: (left) The AI always acts under the safety-critical game policy $(\pi_{\mathbb{O}}^{\text{AI}}, \pi_{\dagger}^{\text{H}})$, making it safe but conservative. (right) The filtered AI uses task policy $\pi_{\boxtimes}^{\text{AI}}$ as long as in the *future* it can apply $\pi_{\mathbb{O}}^{\text{AI}}$ against $\pi_{\dagger}^{\text{H}}$.

The general formulation in Section 4 enables AI systems to preempt potential pitfalls within a specified ODD, but the resulting policy is *only* concerned with safety. If we were to leave $\pi_{\mathbb{O}}^{\text{AI}}$ in control of the AI's entire behavior, as illustrated on the left of Figure 4, it would surely be safe but likely overcautious and unable to provide value to its users. In reality, it is not enough for the AI system to avoid causing failures (if so, we could simply not turn it on), but rather it must do so while assisting its users and performing requested tasks (which may or may not be themselves related to safety). Ideally, we want to *minimally* override the AI's task-driven actions with the safety policy, only intervening at the last possible moment. How can we do this?

The systematic detect-and-avoid functionality we seek closely mirrors the *safety filter* mechanisms established in robotics and control systems, which we reviewed in Section 3. Rather than reinvent a suitable mechanism for human–AI systems, we argue for a frontier framework that builds upon the fundamental principles of safety filtering and extends them to the general interaction problem formalized in Section 4.

Formally, a human–AI safety filter is a tuple $(\pi_{\mathbb{O}}^{\text{AI}}, \Delta, \phi)$ containing:

- fallback policy: $\pi_{\mathbb{O}}^{\text{AI}} \colon \mathcal{Z}^{\text{AI}} \to \mathcal{A}^{\text{AI}}$, aims only to avoid catastrophic failures, without regard to task performance, and is therefore kept as a last resort.

- safety monitor: $\Delta \colon \mathcal{Z}^{\text{AI}} \times \mathcal{A}^{\text{AI}} \to \mathbb{R}$, checks if the fallback $\pi_{\mathbb{O}}^{\text{AI}}$ would still maintain safety *after* a candidate action $a^{\text{AI}}$ is taken from $z^{\text{AI}}$, outputting a positive or negative value following Equation 4.

- intervention scheme: $\phi \colon \mathcal{Z}^{\text{AI}} \times \mathcal{A}^{\text{AI}} \to \mathcal{A}^{\text{AI}}$, permits a candidate action $a^{\text{AI}}$ if if passes the monitoring check and otherwise replaces it with an alternative action that does, for example $\pi_{\mathbb{O}}^{\text{AI}}(z^{\text{AI}})$.

This definition can encompass a broad spectrum of potential future supervisory mechanisms (Legg, 2023) and allows us to construct a new central theorem to understand their guarantees and assumptions.

**Theorem 1 (General Human–AI Safety Filter)** *Consider a human–AI system with AI world model $f^{AI}(z^{AI}, a^{AI}, a^{H}, o^{AI})$ and a safety filter $(\pi_{\mathbb{O}}^{AI}, \Delta, \phi)$. If the AI agent is deployed with an initial internal state $z_0^{AI} \in \mathcal{Z}^{AI}$ deemed safe by the safety monitor under the fallback policy, i.e., $\Delta\left(z_0^{AI}, \pi_{\mathbb{O}}^{AI}(z_0^{AI})\right) \geq 0$, then the interaction under filtered dynamics $f^{AI}\left(z^{AI}, \phi(z^{AI}, a^{AI}), a^{H}, o^{AI}\right)$ with any AI task policy $\pi_{\boxtimes}^{AI} : z^{AI} \mapsto a^{AI}$ and any realization of human behavior satisfying $a^{H} \in \hat{\mathcal{A}}^{H}(z^{AI})$ maintains the safety condition $\forall t \geq 0, z^{AI} \notin \mathcal{F}^{AI}$.*

To date, the concept of a safety filter has only been instantiated for embodied systems with physics-based state spaces (low-dimensional vectors of physical quantities like positions or velocities, governed by well-understood dynamical laws). Here, we are the first to generalize the scope of this mathematical formalism to the much broader context of AI safety. This result lays the theoretical foundations for the algorithmic application of safety filters to *general* human–AI systems, which evolve "latent state spaces" and encode harder to model interactions such as dialogue between a human user and an AI chatbot.

An important aspect of Theorem 1 is that it holds for an arbitrary fallback policy $\pi_{\mathbb{O}}^{\text{AI}}$: as long as the safety monitor $\Delta$ can accurately predict whether $\pi_{\mathbb{O}}^{\text{AI}}$ will succeed at maintaining safety in the future, the intervention scheme $\phi$ can prevent actions that would lead to a vulnerable state, i.e. a state outside the *fallback-safe* set $\Omega^{\mathbb{O}}$. Naturally, if the available fallback policy is not very effective, the filter will be forced to intervene often, restricting the human–AI interactions to remain inside a smaller set $\Omega^{\mathbb{O}}$. This is where the safety game from Section 4 comes in.

**The Perfect Human–AI Safety Filter.** The safety-critical human–AI game we posed in Section 4 implicitly encodes the *least-restrictive* safety filter possible: one that allows maximal freedom to the AI's task policy $\pi_{\boxdot}^{\mathtt{AI}}$ while preempting all future safety failures under the AI's uncertainty. In particular, if we had access to the exact solution to this safety game, such a *perfect* safety filter could be implemented by choosing fallback policy $\pi_{\pmb{\mathbb{O}}}^{\mathtt{AI}}$, safety monitor $\Delta := Q(\cdot, \cdot, \pi_{\dagger}^{\mathtt{H}})$, and switch-type intervention scheme $\phi := \mathbb{1}_{\{\Delta > 0\}} \cdot \pi_{\boxdot}^{\mathtt{AI}} + \mathbb{1}_{\{\Delta \leq 0\}} \cdot \pi_{\pmb{\mathbb{O}}}^{\mathtt{AI}}$.

**Algorithmic Human–AI Safety Filtering.** We conclude by giving a concrete account of how one could practically instantiate a human–AI safety filter for a generative AI model, visualized in Figure 5. Following the common neural network architecture of generative AI models, let **base AI model** (given to us for analysis and safe integration) be comprised of an encoder $\mathcal{E}$ and a decoder $\pi_{\boxdot}^{\mathtt{AI}}$; this decoder is precisely what we have been referring to as the *task policy*, mapping an internal (latent) state $z^{\mathtt{AI}}$ to a proposed output action $a^{\mathtt{AI}}$.

The purple block in Figure 5 depicts the **safety filter** components: the fallback policy, safety monitor, and intervention scheme. Computationally, adversarial reach-avoid RL can be used to obtain an *approximation* of the optimal fallback policy $\pi_{\pmb{\mathbb{O}}}^{\mathtt{AI}}$ from the safety game in Equation 4. A reliable safety monitor $\Delta$ can be implemented by either directly evaluating the learned safety value function at any information state $z^{\mathtt{AI}}$ (safety critic) or by simulating a family of pessimistic interaction scenarios by querying the learned *virtual adversary* $\pi_{\dagger}^{\mathtt{H}}$. In turn, the intervention scheme can range from a simple binary switch (at each time, apply $\pi_{\boxdot}^{\mathtt{AI}}$ if deemed safe, else apply $\pi_{\pmb{\mathbb{O}}}^{\mathtt{AI}}$) to a more sophisticated override (e.g., find a minimally disruptive deviation from $\pi_{\boxdot}^{\mathtt{AI}}$ that is deemed safe).

Figure 5: **Human–AI Safety Filter**. The base AI model encodes the AI's observations into its latent state $z^{\mathtt{AI}}$ which is used as input for its task policy ($\pi_{\boxdot}^{\mathtt{AI}}$). A safety filter includes a learned AI safety strategy $\pi_{\pmb{\mathbb{O}}}^{\mathtt{AI}}$, a safety monitor $\Delta$ that predicts safety risks, and a predictive human model containing a virtual adversary $\pi_{\dagger}^{\mathtt{H}}$ that generates pessimistic predictions of human interaction. Based on $\Delta$, the AI's outputs to the human are filtered by the intervention scheme $\phi$, and modified to guarantee safety.

Even though the components of the safety filter would be approximate by their learning-based nature, the scheme can be leveraged in combination with modern statistical generalization theory, such as PAC-Bayes theory (McAllester, 2003; Majumdar et al., 2020), adversarial conformal prediction (Gibbs & Candes, 2021; Bastani et al., 2022), and scenario optimization (Schildbach et al., 2014; Lin & Bansal, 2023), to maintain a high-confidence guarantee that the AI system will robustly enforce the satisfaction of the human's critical needs throughout the interaction for *all* human behaviors allowed by the operational assumptions. We emphasize that a key strength of this safety framework is that it naturally scales with the rapidly advancing *capability* of modern AI systems: as future generations of language models, vision-language systems, and general AI agents become ever stronger, so will the safety assurances that can be provided through the proposed techniques and system architecture.

# 6 Conclusion

In this paper, we aim to inspire the genesis of a new human–AI safety research community. We take concrete steps towards this by identifying a fundamental synergy between the principled safety formalism offered by control theory and the general representations learned by internet-trained AI systems. By combining lessons from control and AI, we propose a technical roadmap to guide research efforts towards a safety framework that can reliably anticipate and avoid potential hazards emerging during interaction. We propose a frontier framework called the *human–AI safety filter*, wherein an AI system's task policy is systematically monitored and minimally overriden by a safety policy synthesized via safety-critical adversarial self-play.

**Broader Impact Statement**

We expect that the proposed interdisciplinary safety framework will help catalyze a much needed rapprochement between the AI and control systems communities to develop rigorous safety assurances for dynamic human–AI feedback loops. A significant positive impact may come in the form of the first practical safety

frameworks that can not only keep up with the rapid advances in AI capabilities but actively benefit from them to provide stronger guarantees, ushering in a new generation of advanced AI systems that can be trusted *because* of their sophistication, and not in spite of it.

On the other hand, we also highlight possible pitfalls of our proposed human–AI safety framework. The approximate nature of learning-based generative AI makes it extremely challenging to provide a clear-cut delineation of uncertainty, which will likely limit us to statistical assurances in the foreseeable future. These fall short of the stronger *if–then* certificates that we can aspire to in other engineering domains: hard guarantees establishing that, as long as the system's operational assumptions are met, catastrophic failures are categorically impossible (i.e., a failure can only result from an explicit assumption being violated). Even high-confidence statistical assurances can leave human-centered AI systems open to black swan events with extremely low probability but potentially dramatic consequences.

There is a risk that the improved treatment of human–AI feedback loops developed through the proposed agenda could be repurposed and misused by malicious or reckless actors to construct AI systems that exploit interaction dynamics against the interest of their users, for example by seeking to manipulate their decisions. Even with today's relatively myopic fine-tuning approaches, we see a worrying emergence of unintended (e.g., sycophantic) AI outputs as the system learns to secure positive user responses. Future systems equipped with long-horizon reasoning but *without* a proper safety framework could conceivably seek long-term interaction outcomes serving a third party's agenda at the expense of their users' needs.

We nonetheless remain cautiously optimistic: First, human–AI safety filtering does not require teasing apart the likelihood of various conceivable human behaviors in a given context. Rather, safety-directed predictions robustly consider the set of all such plausible behaviors without distinction, making them harder to exploit for manipulation purposes. Second, the need to consider large prediction sets containing both likely and unlikely outcomes aligns well with the inclusion of underrepresented individual behaviors that do not conform to dominant patterns in the training datasets. Finally, provided that future AI systems are deployed with a cyber-secure dynamical safety mechanism that cannot be removed or altered by unauthorized parties, such a framework would help detect and mitigate emergent and intentional misalignment. Naturally, this will require a process of standardization and regulatory oversight; the first step, however, must be to establish *what assurances are possible*. Ultimately, we expect that technical advances in human–AI safety will inform the conversation between technologists, policymakers, political leaders, and the public at large. A timely conversation that, fortunately, is already ongoing.

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
