# OpenReview forum: "Human–AI Safety: A Descendant of Generative AI and Control Systems Safety"
_TMLR — Rejected by TMLR_

### Review · Reviewer_HGaS · 2024-07-18

**Summary Of Contributions:**

This paper addresses an important problem: how to make the next-generation human-AI systems safe and robust. To solve this problem, shedding light on the safety filter technique established in control systems, this paper proposes a general framework that monitor the action of the AI system and intervene in its action when necessary safety constraints are violated.

**Audience:**

Yes

**Broader Impact Concerns:**

There are no concerns on the ethical implications of this work.

**Claims And Evidence:**

No

**Requested Changes:**

My detailed requested changes are listed in the Weakness part; please refer to it.

**Strengths And Weaknesses:**

## Strengths:

1. The investigated problem of how to enhance the safety of human-AI systems is important and not well-solved in modern AI systems.

2. The idea of using the safety filter technique coming from control systems to monitor and intervene in the action of the AI system is novel and interesting.

## Weaknesses:

1. Though the idea of this paper is convincing, the proposed framework lacks of evidence to prove its effectiveness in practical scenarios. I suggest the authors provide more theoretical or experimental results on the practical performance of the proposed framework.

2. Though Theorem 1 provides safety guarantees on the output of AI, how to ensure the output of AI is not overcautious and valuable?

3. One minor issue: the first 'if' is not needed in the sentence 'if if passes the monitoring check	and allow us to ...' on page 10.

---

### Review · Reviewer_Exyn · 2024-08-01

**Summary Of Contributions:**

In this paper, the authors propose combining AI and control theory to enhance safety in human-AI interactions. Human-AI safety filter is proposed to ensure AI systems’ actions remain safe during interactions with humans. Besides, the paper outlines a roadmap using game theory to model interactions and mitigate potential hazards. The conditions and assumptions under which these systems can be expected to operate safely are discussed as well.

**Audience:**

Yes

**Broader Impact Concerns:**

The paper identifies several broader impact concerns associated with the proposed human-AI safety framework in the Broader Impact Statement section.

**Claims And Evidence:**

No

**Requested Changes:**

Please refer to the weaknesses mentioned above.

**Strengths And Weaknesses:**

Strengths:
- The paper presents a novel framework that combines AI and control systems, offering a new perspective on AI safety. It provides an exploration of both AI and control systems' strengths, offering a well-rounded understanding of how these fields can synergize to address safety challenges.
- The proposed roadmap provides clear steps for implementing the proposed safety framework, which could guide future research and development in the field.

Weaknesses:
- The paper lacks specific case studies or simulations demonstrating the framework's practical application and effectiveness. This addition would provide concrete evidence to support the theoretical claims made in the paper.
- The proposed framework relies on operational assumptions about human behaviour that may not always hold true, potentially limiting its effectiveness in certain scenarios. It would be better to provide a more detailed discussion on the assumptions made about the human behaviour.

---

### Review · Reviewer_XDR2 · 2024-09-13

**Summary Of Contributions:**

The paper make a discussion over how to construct a human-AI feedback loop, and proposes
a roadmap towards next-generation human-centered AI safety.

**Audience:**

Yes

**Broader Impact Concerns:**

N / A

**Claims And Evidence:**

Yes

**Requested Changes:**

See Weaknesses.

**Strengths And Weaknesses:**

* Strengths

The discussion is insteresting, which gives us a concrete introduction about how to exploit human-AI interaction to improve the AI safety.

* Weaknesses

The paper may be a good popular science article. It only discusses solutions and technical routes, but does not contain any experiments / evidence. I am not an expert in AI ethics. However, from a technical point of view, I do not think this article is publishable. I will explain a bit more as follows.

1. The authors claim that they propose "a concrete technical roadmap towards next-generation human-centered AI safety", which is mainly described in Section 4 & 5.  As a roadmap, several thereom and formulations are formally proposed without any proof theoretically or experimenatally. Maybe this can only be called a hypothesis？

2. AI-safety is a large topic and the paper tries to cover too much like a survey. However I don't think this is a survey and the author doesn't claim it is a survey. I suggest the author revise the paper to make the claims more focused, clear and easier to understand.

---

### Decision · Action_Editor_Ccut · 2024-11-15

**Recommendation:** Reject

**Comment:**

The reviewers found the paper lacking in theoretical and experimental evidence supporting its claims. In particular, the paper states a theorem on general human-AI safety filters (Theorem 1). However, no proof is given. There are also no experiments or case studies that justify the framework's practical realizability. The paper would also benefit from a deeper discussion of the assumptions the framework makes about human behavior.

The basic issue here is that the paper is fundamentally a position paper. However, TMLR does not have an explicit position paper track. TMLR does accept "surveys that draw new connections, highlight trends, and suggest new problems in an area", but this paper is not quite a survey. I encourage the authors to review the paper based on the reviewers' feedback and to submit to a venue that specifically targets position papers.

**Audience:**

The paper is on AI safety, a key topic in AI and machine learning research that is of interest to a substantial segment of the TMLR community.

**Claims And Evidence:**

This position paper connects the distinct fields of AI safety and control systems safety and argues for an overarching framework of human-AI safety. The main argument is that to provide safety assurances about AI systems, one must reason about feedback loops between AI systems and human users. Based on this argument, the paper gives a unifying formalism for the safety of human-AI systems and gives a technical roadmap for this new area.

**Resubmission Of Major Revision:**

The authors may consider submitting a major revision at a later time.